# Change in hierarchy of the financial networks: A study on firms of an emerging market in Bangladesh

**Mahmudul Islam Rakib[1,2], Md. Jahidul Alam[1,2], Nahid Akter[1], Kamrul Hasan Tuhin[3], Ashadun Nobi [ID][1] ***

**1** Department of Computer Science and Telecommunication Engineering, Noakhali Science and Technology University, Sonapur, Noakhali, Bangladesh, **2** Department of Computer Science and Engineering, Daffodil International University, Ashulia, Dhaka, Bangladesh, **3** Department of Computer Science and Engineering, Z.H. Sikder University of Science and Technology, Shariatpur, Bangladesh

* ashadunnobi_305@yahoo.com

**Data Availability Statement:** The dataset containing the prices of the companies included in the DSE index is publicly available on the website https://www.investing.com and was also retrieved from there for use in this research.

## Abstract

We investigate the hierarchical structure of Dhaka stocks' financial networks, known as an emerging market, from 2008 to 2020. To do so, we determine correlations from the returns of the firms over a one-year time window. Then, we construct a minimum spanning tree (MST) from correlations and calculate the hierarchy of the tree using the hierarchical path. We find that during the unprecedented crisis in 2010–11, the hierarchy of this emerging market did not sharply increase like in developed markets, implying the absence of a compact cluster in the center of the tree. Noticeably, the hierarchy fell before the big crashes in the Bangladeshi local market, and the lowest value was found in 2010, just before the 2011 Bangladesh market scam. We also observe a lower hierarchical MST during COVID-19, which implies that the network is fragile and vulnerable to financial crises not seen in developed markets. Moreover, the volatility in the topological indicators of the MST indicates that the network is adequately responding to crises and that the firms that play an important role in the market during our analysis periods are financial, particularly the insurance companies. We notice that the largest degrees are minimal compared to the total number of nodes in the tree, implying that the network nodes are somewhat locally compact rather than globally centrally coupled. For this random structure of the emerging market, the network properties do not properly reflect the hierarchy, especially during crises. Identifying hierarchies, topological indicators, and significant firms will be useful for understanding the movement of an emerging market like Dhaka Stock exchange (DSE), which will be useful for policymakers to develop the market.

## 1. Introduction

The stock market in Bangladesh has experienced significant growth in recent years, establishing itself as a prominent participant in the South Asian region. CEIC Data reports that the market capitalization of the Bangladesh stock market reached $41.241 billion in 2023, which is

**Funding:** This research was supported by the ICT Division of Bangladesh [The grant number is 19FS32950].

**Competing interests:** The authors have declared that no competing interests exist.

approximately 10.2% of the country's GDP [1]. The market has experienced a rise in foreign investors, with the count of foreign portfolio investors (FPIs) surging from 200 in 2010 to more than 1,000 in 2022 [2]. The market has been garnering the attention of international investors and exhibiting strong performance in recent years. Many studies have been done on developed and developing financial markets [3–6]. They did statistical and network analyses on price fluctuations to extract information from the market. In particular, the topological properties of the Russian stock market's network generated from the correlation coefficients are compared with those of the US and China [3]. Another study applied partial information decomposition to some developed American and European stock markets to identify the amount of synergistic information transfer to some developing Asian markets and concluded the US as the most influential and four developing Asian markets, Korea, Tokyo, Hong Kong, and Singapore, as the most influenced markets [4]. The feature ranking method has been implemented to analyze the topological properties of the threshold networks of a developed American market and 21 developed and developing world indices [5, 6]. Some research investigated the structural change and dynamics of the state of the emerging stock markets, such as the PSX of Pakistan [7] and the DSE of Bangladesh [8], using different correlation and network techniques. Dhaka Stock Exchange (DSE) is an emerging market, and fluctuations in its time series are rather interesting. The market is affected by different types of crises and consequently, the DSE index fluctuates over time [8]. The biggest crisis in this market is the 2011 Bangladesh market scam. Before the event, the global financial crisis in 2008, which affected the majority of the financial sectors, had seen to have a less significant impact on the market. But after the scam, the market reflects different global financial events, like the 2015–16 market selloff, the beginning of the trade war in 2018, and the COVID-19 pandemic in 2020. This is evident in the globalization of the market, and to study this transformation, we analyze the market for the time period 2008 to 2020.

The aim of this study is to analyze the time series of an emerging market from 2008 to 2020, covering different crises using a statistical and complex network approach and to compare the hierarchical structure of this market with other emergent, developing and developed market done in last decades. We construct the Minimum Spanning Trees (MSTs) from cross-correlations of price returns between 110 stocks on the Dhaka Stock Exchange with a yearly time window. The topological hierarchy F is measured using the hierarchical path concept. The hierarchy of financial networks has been quantified with the evolution of time. We identify periods when the hierarchical transition occurs. It is noticeable that the hierarchy score falls sharply before different big crashes in the Bangladeshi local market, which is useful in forecasting any upcoming big crisis. To complete the analysis of the hierarchical structure of the network, we also refer to the analysis of other topological indicators. The contribution of our study is that we investigate the hierarchical structure and organization of an emerging financial market, e.g., the DSE of Bangladesh in this research, for the first time, which might help the authorities and policymakers of this and similar markets to identify behavioral differences from that of developed and developing markets that can be used in promoting them to the next higher levels. We showed significant differences in the hierarchical structure of the tree of an emerging market during different crises than those of developed and developing markets, which had not been investigated before.

The article is organized as follows: In Section 2, some similar methods in the literature are discussed, and the motivation for using our methods is explained. Section 3 describes the data and main statistical and graph theoretical tools used in this research briefly. Our findings are depicted and analyzed in Section 4. Section 5 concludes the paper with a summary of our main findings.

## 2. Literature review

In the last decades, different methods and techniques have been used to analyze financial market data [1–28]. Network theory has been extensively used in financial data analysis, in particular, the MST. The MST is one of the essential network reconstruction techniques, which Mantegna first applied to the correlations of stock returns and found that similar types of companies are clustered in the tree [9]. Since then, the MST has been applied to numerous financial time series of local and global markets to understand the market clustering structure over time [10–12]. The correlation-based MST of 1071 stocks of NYSE was constructed, and it was observed that sectorial companies tend to cluster together for a local market [10]. For the global market, it was shown that world indices are grouped on the MST according to their geographical location [11]. A modified MST technique was applied to absolute cross-correlation coefficients of 28 stock markets, 21 currencies, and 20 commodities, and the study showed that similar types of financial assets tend to have stable connections in the MST [12]. Another category of articles used MST to monitor the structural change of the network due to financial shock [13, 14]. The structural change of the MST due to crises is also found in the local market. A star-like tree with a super hub during the global financial crisis was found in the German stock market FSE [13] and the Korean stock market KOSPI [14]. However, before and after the crisis, the tree was chain-like and decorated by many hubs. Some research observed topological properties of the MST to extract the financial market's evolutional information [15–17]. An associative analysis of the market volatility and MST's network properties using APL, maximum degree, and BC was performed to show the effect of volatility on the market [15]. MST has been applied to model many other complex market structures [16, 17]. The PMFG was proposed as an alternative tree representation tool of the MST, and as an application, it was tested on 100 stocks of the NYSE [18]. Another study constructed the PMFG for 143 stock indices of 59 different countries and confirmed that globalization has been increasing in this market since 2000 [19]. However, one advantage of the MST is that it provides a parsimonious representation of the network with $N - 1$ edges of the tree out of all possible $N(N - 1)/2$ interconnectedness, retaining the essential characteristics of the links in the group [20]. It also represents the network in a way that allows the identification of the central stocks that tend to be the important indices and the peripheral or less influential ones in the market. The diameter of the tree indicates the level of market integration and interactions. The dynamics of the state of the market directly influence the structure of the MST. Moreover, MST doesn't require special assumptions. For these, the MST is a very effective tool in financial market analysis. Besides, MST depicts the hierarchical structure of a financial market, which is useful in analyzing hierarchical alteration in that market and is one of this article's aims.

In financial market, the MST is constructed from correlation, partial correlation, mutual information, etc. [9–15, 21, 22]. The MST is a hierarchical structure, and its hierarchy score can be measured. Considering the degrees of the nodes in the network, the measure of the hierarchy F, known as the hierarchical path, is introduced by Trusina [23]. The F is the fraction of paths between all pairs of nodes that are degree hierarchical. Nobi et al. adopted the concept of the hierarchical path and quantified the topological hierarchy F of the MSTs constructed from cross-correlations between stock indices of the S&P 500 to show how the hierarchical organization of the financial network has been shaped with the evolution of time and also due to the effect of financial crises [24]. Another article investigated the hierarchical organization of MST of the world trade market using the hierarchical path concept [25]. The higher values of the F imply that the high-degree nodes are placed in the center [24, 25]. Besides the hierarchical path, some other hierarchical measures, like agglomerative hierarchical clustering on KOSPI correlations [22], hierarchical clustering on the world trade and stock market using

cophenetic correlations coefficient (CCC) [26, 27], hierarchical measure using allometric scaling on trade of ten commodities [28], were applied to financial systems. Agglomerative hierarchical clustering is a simplest technique to cluster the firms. Using this technique, financial events are not identified clearly [28]. In CCC, choosing the proper hierarchical clustering algorithm and number of clusters is always a key question for researcher. The reason behind this could be that different cluster validation and comparison techniques give contradictory results in most cases. The allometric scaling is applied in trade flow network to show hierarchical organization of different commodities [28]. The scaling exponent is used to show the dissimilar organization of different products. If one considers error bar, it is difficult to distinguish the products using the exponents.

However, the hierarchical path concept that was used to measure the hierarchy score counts all possible hierarchical paths while calculating. It also considers the position of the high-degree nodes in the network, and a higher score implies the existence of a few high-degree nodes placed around the center. These properties are effective in modeling complex systems like the financial market.

## 3. Data and methods

### 3.1 Daily return and correlation

We use the daily closing prices of 110 companies of the DSE for the time period from January 1, 2008, to December 31, 2020. These 110 companies existed on the market throughout this whole time period. We calculate the log-returns of the stock prices of each company for a one-year time window as follows

$$r_i(t) = [ln\, P_i(t) - ln\, P_i(t-1)] \tag{1}$$

Where $P_i(t)$ is the daily closing price of index $i$ at time $t$ and $\sigma_i$ is the standard deviation of the whole time series for that index. Then, the equal time cross-correlation at a one-year time window $T$ (approximately 250 days) is calculated from the normalized return between two indices, $r_i$ and $r_j$ by

$$C(i,j) = \frac{\left\langle r_i(t) * r_j(t) \right\rangle - \langle r_i(t) \rangle * \langle r_j(t) \rangle}{\sigma_i * \sigma_j} \tag{2}$$

Where $C(i,j)$ represents the cross-correlation matrix between stock $i$ and stock $j$. $\langle \bullet \rangle$ represents the mean and $\sigma_{\bullet}$ represents the standard deviation of stock returns, with $\langle r_i(t) \rangle = \frac{1}{T}\Sigma_{t=1}^{T} r_i(t)$ and $\sigma_i = \sqrt{\frac{1}{T}\Sigma_{t=1}^{T}[r_i(t) - \langle r_i(t) \rangle]^2}$

### 3.2 Network construction

The MST is a well-known technique to visualize the organization of the nodes in a loopless subgraph consisting of $(N-1)$ links reaching all $N$ nodes. It is an efficient approach to capturing the most relevant correlations for each node. The MST is constructed by calculating the distance matrix of the indices [24, 29]. The distance matrix is defined as

$$d_{ij} = \sqrt{2(1 - C_{ij})} \tag{3}$$

Where $d_{ij} = 0$ if the price time series of the index $i$ and $j$ are perfectly correlated, and $d_{ij} = 2$ if the price time series of the index $i$ and $j$ are perfectly anti-correlated. The MST has been built

following Kruskal's algorithm to find the $N - 1$ most important correlated pairs of the indices among the $N(N - 1)/2$ possible pairs.

## 3.3 Measure of hierarchy F

The concept of a hierarchical path [23] is applied to quantify the hierarchy of financial networks. A single path between two nodes is hierarchical if any of the following conditions are true.

- It consists of an 'up path' where the degrees of each two consecutive nodes $k_i$, $k_j$ along the path satisfy $k_i \leq k_j$, and then followed by a 'down path' where the consecutive nodes have lower or equal degrees

- It consists of an 'up path' alone

- It consists of a 'down path' alone

In simple terms, the degrees of the nodes along a degree-hierarchical path are either ascending or descending or first ascending and then descending in order. If there are two or more paths between nodes $i$ and $j$, each path that satisfies one of the above three conditions will be counted as hierarchical. There is no hierarchy for multiple combinations of consecutive up and down paths. For instance, up-down-up and down-up-down are not hierarchical. The amount of the hierarchy F is defined by the fraction of paths in the network that are degree-hierarchical [23, 25].

$$F = N_H \bigg/ \frac{N(N - 1)}{2} \tag{4}$$

Where $N_H$ is the number of available hierarchical pathways in the MST.

## 3.4 Topological properties

**3.4.1 Node degree.** In a network, the degree of a node is the number of connections that it has to other nodes. If $e_{xy}$ is an undirected edge between node $x$ and node $y$ then the degree of node $x$ can be defined as [30],

$$k_x = \sum_{y \neq x} e_{xy} \tag{5}$$

**3.4.2 Betweenness centrality.** Betweenness Centrality of a node $v$ is the sum of the fraction of all-pairs shortest paths that pass through $v$ [5].

$$C_B(v) = \sum_{s,t \in V} \frac{\sigma(s, t|v)}{\sigma(s, t)} \tag{6}$$

Where $V$ is the set of nodes, $\sigma(s,t)$ is the number of shortest $(s,t)$-paths, and $\sigma(s,t|v)$ is the number of those paths passing through some node $v$ other than $s$, $t$. If $s = t$, $\sigma(s,t) = 1$, and if $v \in s,t$, $\sigma(s,t|v) = 0$.

**3.4.3 Network diameter.** The diameter of graph G is the maximum eccentricity which is [31],

$$D = \max_{v \in V} e_v \tag{7}$$

where $V$ is the set of nodes in G, $e_v$ is the eccentricity of node $v$, which is the maximum distance from $v$ to all other nodes in G.

**3.4.4 Average path length.** The average shortest path length is the average of the shortest path lengths between every pair of nodes in the graph. The average shortest path length is [6],

$$a = \sum_{s,t \in V} \frac{d(s,t)}{n(n-1)} \tag{8}$$

where $d(s,t)$ is the shortest path from $s$ to $t$, and $n$ is the number of nodes in G.

# 4. Results and discussion

The daily closing price of 110 firms of the DSE is monitored from 2008 to 2020, which covers different local and global financial crises. In Fig 1a, we show the logarithmic prices of four firms of the DSE, where two are hub companies and the others are peripheral in the tree. We

(a)
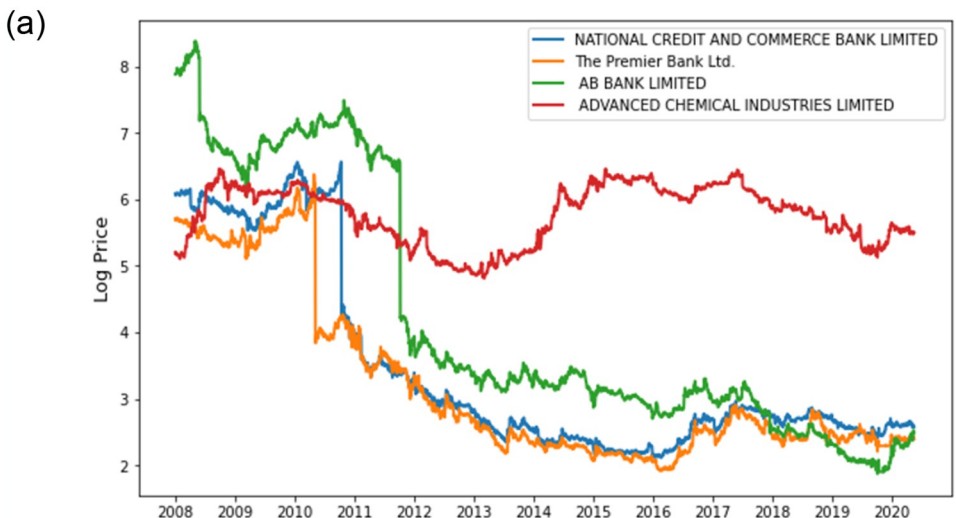

(b)
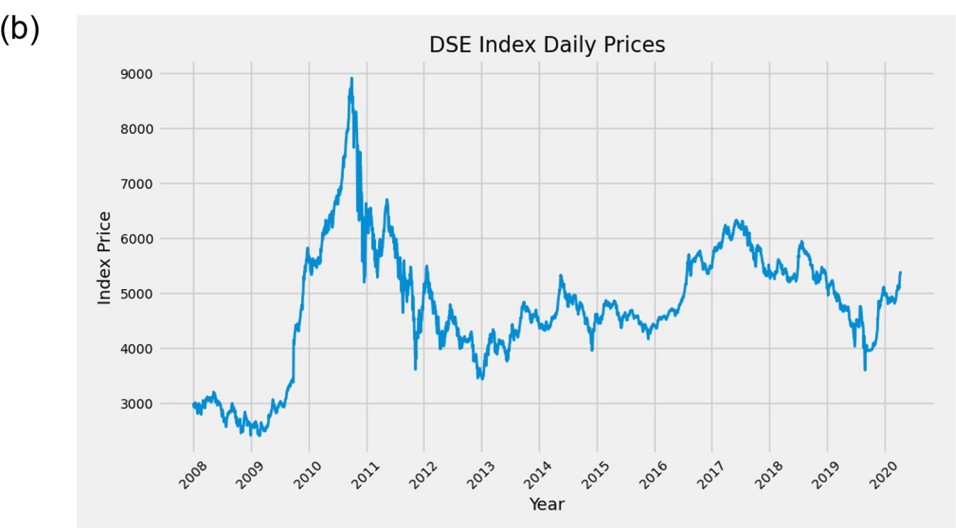

**Fig 1. Logarithmic values of prices of the four companies and the index price of the DSE.** (a) Logarithmic price of four companies (b) Index price of the DSE. National Credit and Commerce Bank Ltd., and The Premier Bank Ltd. are the hub companies, and AB Bank Ltd., and Advanced Chemical Industries Ltd. are the peripheral companies in the network in 2008 and 2020, respectively.

observe the typical trend in stock prices for the hub and the peripheral. We identified nodes as hub and peripheral based on the extent of their connectivity. The node with the highest degree in the tree is referred to as the hub node, while the nodes with the lower degree that lie outskirt in the tree are termed as peripheral nodes. The National Credit and Commerce Bank Limited and The Premier Bank Limited were two hubs during the years 2008 and 2020, respectively. At the same time, two peripheral companies in the tree were AB Bank Limited and Advance Chemical Industries Limited for the periods 2008 and 2020, respectively. Before the financial market crash of 2010–2011, most of the company's prices were overestimated than regular prices, the reason behind this is unknown. The companies whose prices are not overestimated, like AB Bank Limited, did not go down during the crisis of 2010–2011. Even though AB Bank stayed out of the financial crisis in 2011, the company's price dropped in 2012 because of loan fraud and the impact of a sponsor director. ACI's pricing remained consistent and distinct from that of other companies in the DSE due to its diverse business operations across multiple sectors, good budgeting, cost control, and strategic financial planning. However, the prices of all four firms exhibited an upward trend with fluctuations during 2016–17, just after the market selloff and before 2018, the most hierarchical year in our monitored time span, discussed in one of the later sections. During the COVID-19 pandemic in 2020, all the monitored companies except The Premier Bank showed an uptrend, although the DSE index experienced a sharp fall in March and stayed there till June due to the epidemic [32]. We also illustrate the index price of the DSE in Fig 1b and find similar trends with Fig 1a during different periods, especially the downturn around 2011. Here, we conclude that the financial crisis doesn't reflect its influence on the price of all firms, and most of the stocks of the DSE haven't yet recovered from their sharp fall during the 2011 Bangladesh market scam.

Fig 2 depicts the probability distribution function of the cross-correlations of stocks for the two years 2010 and 2018, with 2010 having the lowest hierarchical score and 2018 having the greatest. Both curves deviate from a bell-shaped Gaussian distribution. The central values of the distribution curves are located on the positive side, which means the majority of the correlation coefficients in the range are positive. The distributions show positive skewness with the right tail, whereas the curve of 2010 shows a longer right tail. This indicates that during 2018, when the DSE index fell by 13.75%, there was a greater possibility of larger positive correlations between stocks. The kurtosis of the correlation distribution during 2010 was higher, implying less variance, which means the majority of correlation coefficients between stocks

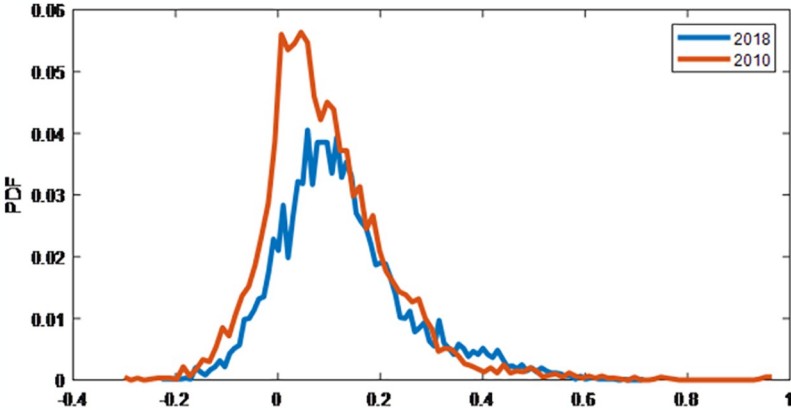

**Fig 2. The probability distribution function (PDF) of cross-correlation coefficients between stocks during 2010 and 2018.** Both curves deviate from the Gaussian distribution. At the beginning of the ESD crisis, the PDF shows a narrow distribution, while the curve during 2018 is wider.

were constrained between 0 and 0.1 at the beginning of the European sovereign debt (ESD) crisis. The probability distribution function of the correlation during 2010 was leptokurtic, while the distribution curve for 2018 showed more variance, and a greater correlation between stocks was more probable. So, in 2010, the correlation coefficients were closer to each other, making the kurtosis higher, while the PDF showed greater variance in 2018, indicating a greater number of higher positive correlations were possible during this period. By relating the probability distribution with hierarchy, we notice that the correlation coefficients are far from each other in a more hierarchical network, showing high variance and indicating that higher positive correlations are more probable; hence, the kurtosis of the PDF is lower.

## 4.1 Structural dynamics of spanning tree

The Minimum Spanning Trees (MSTs) constructed from the correlations of 110 companies in the DSE are shown in Fig 3. Here, we generate the network structures from the correlation coefficients for 2010 and 2018 using the Kruskal algorithm. In Fig 3a, we can see a tree-like network with deep branches at the beginning of the local market crash in 2010. At the end of 2010 and the beginning of 2011, DSE was affected by a severe crisis known as the 2010–11 Bangladesh share market scam [33]. A hub node with a maximum degree of eight in the tree is located near the center. The tree branches form some loose clusters at the periphery, making the branches chain-like in shape. The tree is the least hierarchical in 2010 among our monitoring time periods due to these chain-like branches. This type of tree structure before a crisis was observed in the S&P 500 market, which consists of 443 stocks in the network with the hub near the center, just before the US subprime crisis [34]. On the other hand, Fig 3b depicts a tree network with dense branches in 2018. The branches of the MST form compact clusters except for one long branch that goes downward in the figure. A hub with a maximum degree of ten in the tree forms a star-like cluster in the periphery. These dense branches make the MST of 2018 the most hierarchical tree in our monitored time span. A similar peripheral star-like cluster with the hub was found to form in the MST of KOSPI 200 during the dot-com bubble crisis, where the tree was also peripherally compact [35, 36]. Although the depths of both

(a)　　　2010　　　　　　　　　　　(b)　　　2018

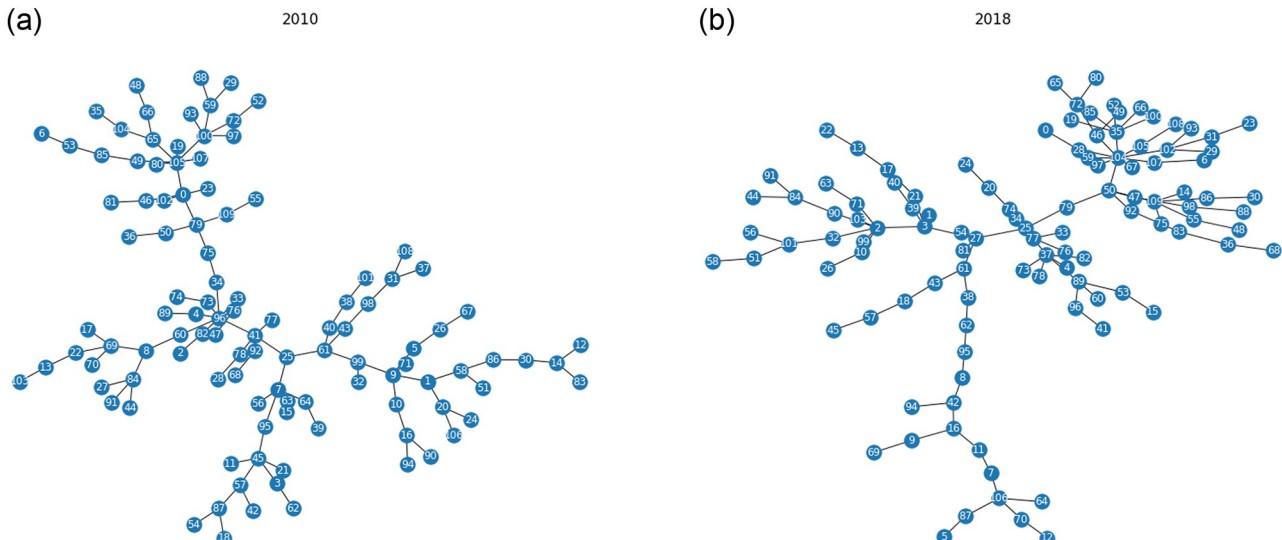

**Fig 3. The minimum spanning trees (MSTs) of the DSE (a) during 2010 and (b) during 2018.** The MST during 2010 was the least hierarchical, and during 2018 it was the most hierarchical among our monitoring time periods. Node-label to company name mapping is given in S1 Appendix.

MSTs in Fig 3 are the same, more nodes are linked by the hub nodes in the tree constructed from the correlations of stocks during 2018. We observe a dissimilarity in the transitions of the MSTs in the German [13] and Korean [14] stock markets, where the hub shifted from the periphery to the center and formed a superhub at the center during crises, which is opposite from the emerging market. To complete the analysis of the topology of MSTs' network structures, different topological indicators, network degree, BC, APL, and tree diameter, are observed in the following sub-sections.

## 4.2 Hierarchical topology

Following the methods of the hierarchical path, we measure the hierarchy score F of the MST constructed from the correlations. We observe the general trend of the F between the years 2008 and 2020 when the DSE market faces different crises, as shown in Fig 4. The hierarchy in an emerging market like DSE is much lower than in developed markets, which can be realized by comparing it with the F of a developed market like USA. It implies that there is no exclusively influential company in the market. S4 Fig in the S2 Appendix illustrates how network hierarchy in the US market changes from 2008 to 2020. According to S4 Fig in the S2 Appendix, the hierarchy of the US developed market had a significant increase during the financial crises of 2011 and 2015. Although the hierarchy was comparatively lower during the financial crisis in 2008 and the COVID-19 pandemic in 2020, yet they remain significantly higher than DSE. A superstar-like hierarchical network structure of the German stock market with a giant hub in the center is also noticed during a market crash [13]. Similarly, in the developing stock market of Korea, nodes are centrally connected with a superhub, forming a hierarchical network during a crisis [14]. However, the MST of the Pakistan stock market, which is an emerging market, was non-hierarchical and chain-like during the global financial crisis [7]. A similar hierarchical structure is observed in the emerging DSE. For example, a big crash occurred in the DSE around 2010–2011. But the hierarchy does not increase as developed markets in this period. During the COVID-19 pandemic, the network also shows less hierarchical structure. This is due to the absence of a giant hub in the center of the tree. It implies that emerging markets are more vulnerable to crisis shocks than developed markets.

In the hierarchical topology of the DSE, sharp drops are noticed before different crashes in the market. The hierarchy was lowest at the beginning of the biggest crisis of DSE in 2010 and

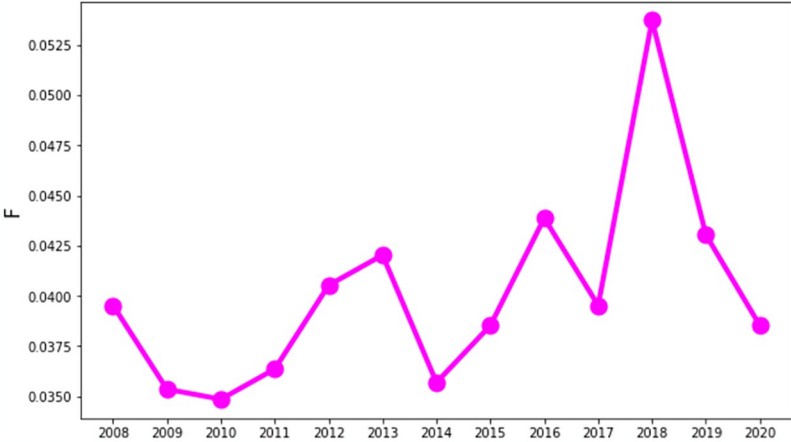

**Fig 4. Change of the hierarchy with the evolution of time for MST.** Sharp drops are observed before different local crises in the Bangladeshi market.

showed a slight uptrend during the market scam in 2011. Similarly, the stock market selloff occurred in 2015–2016, and the hierarchy was lower just before these periods. Another drop was seen in 2017, just before the election year in Bangladesh, which was also the beginning of the world trade war. The hierarchy fell sharply again in 2019 and 2020, implying that the market is in turbulence and that a financial shock is on the way may be as a result of the Black Swan COVID-19 or some other macroeconomic factors. The lower the hierarchy, the higher the financial risk in the near future. We observe a drop in hierarchy before the big local crashes of 2011–2012, 2015–2016, etc. This trend can be used as an indicator and predictor of market turbulence in an emerging market like the DSE. However, we observe the sharp rise of the hierarchy during mild crises such as 2013 due to a sudden drop of the index, 2015–2016 due to the market sell-off, and 2018 when the DSE index fell around 14%. The structure of MSTs for firms in the DSE is less hierarchical than in developed markets [24], which can be understood by comparing the peripherally clustered DSE's MSTs with the centrally coupled MSTs of the S&P 500. The lack of a hierarchical structure implies that the market is fragile. The change in the hierarchy in different periods can be used to identify market movements.

## 4.3 Largest degree companies

Fig 5 depicts the curve of the largest degrees of stocks for each year. We find that the majority of the stocks with the greatest year-wise degree in the network are financial companies. Financial companies were also seen to dominate and act as hubs in different markets around different timelines in the Warsaw Stock Exchange during 2007–2008, where a financial company formed a superhub [37]. On the contrary, industrial companies of USA were perceived as the dominant players and central nodes in various markets at different points in time, as illustrated in the S5 Fig in the S2 Appendix for the S&P 500 from 2008 to 2020. The maximum number of links connected to any node is twelve in Fig 5, which occurred in 2012, 2014, and 2015, and the companies are Federal Insurance Company Ltd., Standard Bank Ltd., and Golden Son Ltd., respectively. The lowest maximum degree was attained by Rupali Insurance Company Ltd. in 2017. Premier Bank Ltd. obtained the maximum number of links in the network twice, in 2018 and 2020. We notice that the largest degrees are minimal compared to the total number of nodes in the tree, implying that the network nodes are somewhat locally compact rather

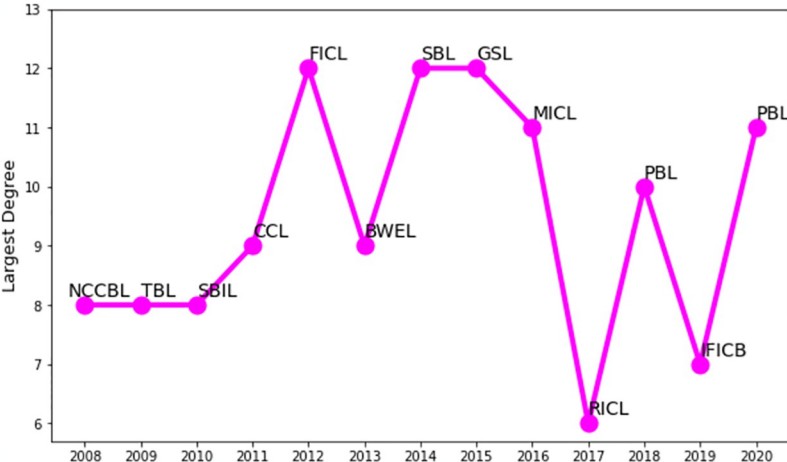

**Fig 5. Year-wise largest degree of stock companies in the DSE network.** The full names of the companies can be found in S1 Appendix.

than globally centrally coupled. Similarly, compact peripheral clusters were observed in the Korean stock market's MSTs, especially after the global financial crisis, where the largest degree was also minimal [14]. For this reason, the DSE network is less hierarchical compared to the developed stock markets [24]. We show the hub (largest degree node) ranking of the DSE companies for two different years in Tables T1 and T2 of the S2 Appendix.

## 4.4 Central companies

Central stocks are important companies in the market. There are many ways and viewpoints of identifying the central nodes in a network. Betweenness Centrality (BC) is such a technique that captures how much a given node is in-between others. The curve of the most betweenness central stocks in each year is shown in Fig 6. We find that the majority of the stocks with the greatest year-wise centrality in the network are financial companies that occur eight times out of thirteen monitored years. We observe that stocks with the largest degrees were also placed in the center of the network in 2010–12 and 2015. Federal Insurance Company Ltd., one of the leading and oldest private general insurance organizations in Bangladesh, was positioned centrally in the network for two consecutive years, 2012 and 2013. City General Insurance Company Limited, another leading non-life insurance company, was located centrally for two different years during our monitored time span. According to the theory of the hierarchical path concept, the most central node should be the hub, and its BC should be higher during a high hierarchical phase of the market. Although for some years, e.g., 2012, 2015–16, and 2018, we find similarities between the trends of BC and hierarchy, reverse changes of patterns are also observed, especially in 2019–20. By comparing with the highest degree nodes, we notice different BC companies in those years. For this behavior, the BC is not always reflected in the degree hierarchy of an emerging market like the DSE. The curve of the BC shows a rising trend during different local crises in the market, indicating centered network structures at the shock of crises, and the most centered network is observed during the COVID-19 pandemic. Based on our findings from the largest degree and centrality measure, we can conclude that financial companies are the important stocks of the Bangladeshi market, whereas insurance companies are the influencing financial institutions. According to PwC [38], the insurance sector has the potential to make a positive contribution to the overall standard of living of the country's populace. The sector can help reduce the burden on the overall spending on

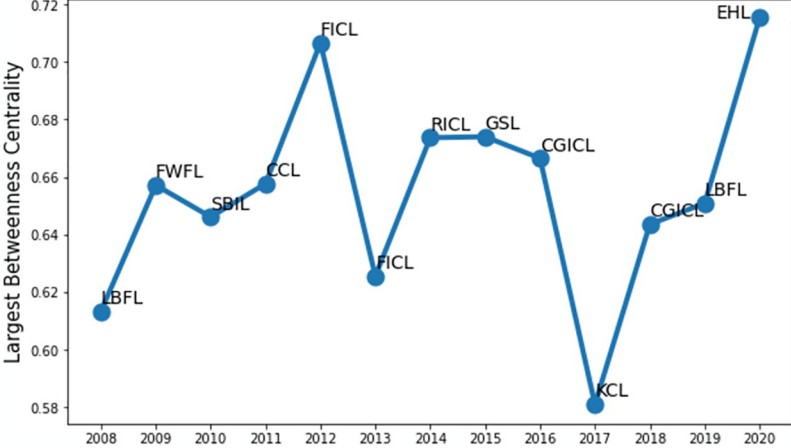

**Fig 6. Annual most betweenness central stock companies in the DSE network.** The full names of the companies can be found in S1 Appendix.

healthcare. The insurance sector can also help manage risks in an economy at a micro level, safeguarding households and companies from a myriad of risks [38]. Although the influence of a company depends on various factors such as P/E (Price-Earnings ratio), EPS (Earnings Per Share), ROA (Return on Assets), SZ (Firm's Size), etc., the average P/E ratio of sample insurance companies in DSE is below 20 (except for 2011) [39], indicating a favorable investment opportunity for small-scale investors. It is also seen that the ROA and EPS of some selected insurance companies have significant positive impacts on market movement. S6 Fig in the S2 Appendix displays the stocks with the highest betweenness centrality in the S&P 500 market from 2008 to 2020. Based on the figure, the companies that have the highest centrality in the network vary from year to year. These stocks are primarily from the financial and industrial sectors. It is already known that financial companies are the most important stocks in some stock markets, e.g., in the Brazilian market with the largest centrality measures [40], and insurance companies directly influence the systemic risk in the European market [16].

## 4.5 Shortest path analysis

The shortest path is a network measure that identifies the closeness of nodes in the network. We calculate the longest shortest path or the diameter of the network and the average shortest path (APL) between any pair of nodes, as shown in Fig 7a and 7b, respectively. It is seen in the literature that the shortest path indicators become lower when the network is hierarchical [24]. In addition, S7a and S7b Fig in the S2 Appendix demonstrate that the shortest path length is significantly lower during periods when the hierarchy is higher for the S&P 500 market. However, for the emerging DSE, the hierarchy and shortest path topologies show some irregular patterns. For example, we observed the lowest shortest path in 2015, which imply that we should have found a higher hierarchy in this period. But instead, we don't find a significant hierarchy score. On the other hand, we find the highest hierarchy in 2018, shown in Fig 4, but a remarkable shortest path is not found then. From Fig 7, it is noticeable that the diameter and APL curves have a somewhat similar trend, which indicates the change of the pair-wise shortest path length in accordance with the diameter of the network. A drop in both curves was found in 2010 at the beginning of the Bangladesh stock market scam. Another drop was observed in 2012, followed by a sharp rise in 2013, specifying the long-lasting instability of the market due to the influence of the local scam during 2010–11. A lower diameter implies a shallower MST, and a smaller APL indicates a tightly coupled network structure. The sharp fall both in diameter and APL was noticed at the beginning of the global market selloff in 2015, implying a highly influenced market structure during this period. Literature has already found that APL responds negatively to market volatility, and the market is more volatile during crises [15]. After 2016, the curves continued to decline except in 2020 for the diameter, which specifies that the network is shrinking, intra-market transactions are increasing, and the structure is getting vulnerable to the rapid spread of financial shock. In 2020, during the COVID-19 pandemic, the APL reached its local minimum, as did the hierarchy shown in Fig 4. This means nodes were randomly linked in the network, making the market vulnerable to the crisis during the pandemic. In summary, the shortest path topologies do not reflect the hierarchy of the emerging market properly, which was found for the developed market [24].

## 5. Conclusion

We study statistics and the dynamic transition of the hierarchy in the financial networks of the Dhaka Stock Exchange for 13 different years. To do so, the minimum spanning tree is constructed from cross-correlations of stocks of the DSE for a one-year time window, and the hierarchical path is applied to measure the hierarchy score F of the financial network. The F

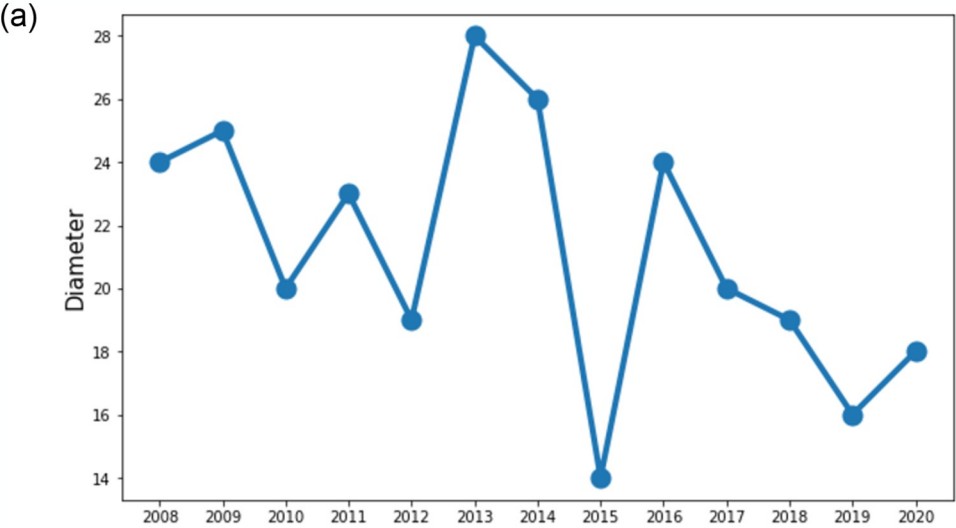

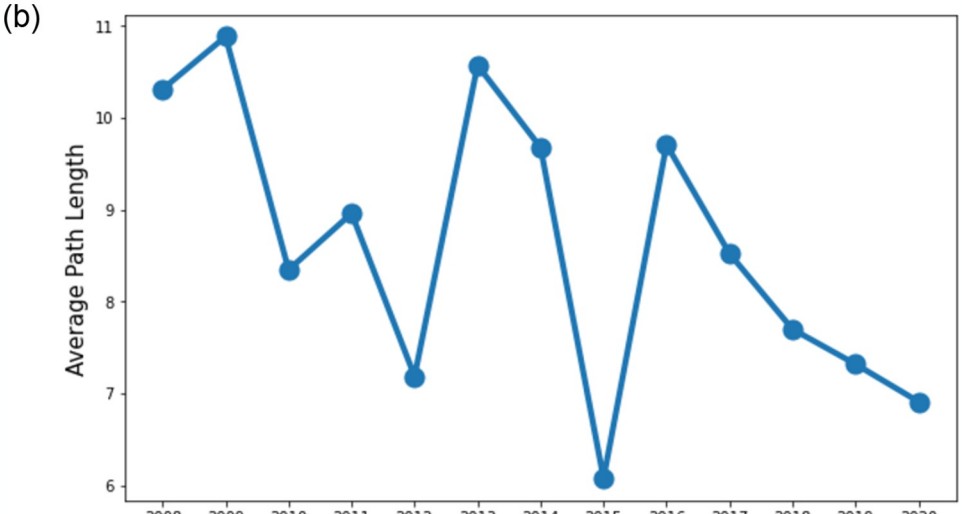

**Fig 7. Topological properties regarding the shortest path of the DSE network.** (a) Network Diameter (b) Average Shortest Path Length.

can identify different financial states and can measure the effectiveness of crises on the financial market. We observe that the hierarchy score falls sharply before different big crashes in the Bangladeshi local market, which is useful in forecasting any upcoming big crisis. When we look at the network structure of developed markets like the S&P500, we find that the market structure is more hierarchical during a crisis. We also observe a similar configuration for developing markets like KOSPI during a crisis. However, when we look at the network structure of emerging markets like DSE, we find that the tree is less hierarchical during big crises like the 2011 scam. Similar behavior is also seen in the Pakistan stock network. We find that the network hub and central nodes with a maximum degree and BC measure are mainly financial companies, more specifically the insurance companies, i.e., they play important roles in the Bangladeshi market. For a poor country like Bangladesh, the influence of the insurance company is remarkable since it has been crucial in the lives of mass people and the related

aspects those are concerned about. From literature, it is found that different factors like P/E, ROA, and EPS of insurance firms are significantly impacting the invest opportunities and market movements of the DSE. These are a few possible factors why insurance companies are influential in DSE. By analyzing different network properties, we observe the degree of the hub node increase during most hierarchical years. But the other properties, especially the shortest path indicators, are not properly reflecting the hierarchical dynamics of the emerging market structure, which was seen to align in developed markets. This indicates the usefulness of the hierarchical topology in analyzing financial systems from a different perspective.

Although, there are different techniques to measure the hierarchy of the tree, the performance of our technique is better than others since we consider all possible hierarchical path of the tree to measure the hierarchy. One drawback of our technique is that it is useful only for the tree. In future, we will construct PMFG for the companies of the market and will measure the hierarchy of the graph using proper hierarchical techniques to extract more information from the market.

## Supporting information

**S1 Appendix. Mapping of company names to company symbols and nodel labels.**
(DOCX)

**S2 Appendix. Supporting figures for comparative analysis and tables for hub ranks.**
(DOCX)

## Acknowledgments

The authors acknowledge the support provided by the Information and Communication Technology Division of Bangladesh.

## Author Contributions

**Conceptualization:** Ashadun Nobi.

**Data curation:** Md. Jahidul Alam.

**Formal analysis:** Mahmudul Islam Rakib, Ashadun Nobi.

**Funding acquisition:** Md. Jahidul Alam.

**Investigation:** Mahmudul Islam Rakib, Nahid Akter, Ashadun Nobi.

**Methodology:** Mahmudul Islam Rakib, Kamrul Hasan Tuhin.

**Software:** Md. Jahidul Alam.

**Supervision:** Ashadun Nobi.

**Validation:** Mahmudul Islam Rakib, Nahid Akter.

**Visualization:** Md. Jahidul Alam, Kamrul Hasan Tuhin.

**Writing – original draft:** Mahmudul Islam Rakib, Ashadun Nobi.

**Writing – review & editing:** Mahmudul Islam Rakib, Nahid Akter, Kamrul Hasan Tuhin, Ashadun Nobi.

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
