## [Decision Letter · Decision Letter 0]

3 Jan 2024

PONE-D-23-35741Change in hierarchy of the financial networks: A study on firms of an emerging market in BangladeshPLOS ONE

Dear Dr. Nobi,

Thank you for submitting your manuscript to PLOS ONE. After careful consideration, we feel that it has merit but does not fully meet PLOS ONE’s publication criteria as it currently stands. Therefore, we invite you to submit a revised version of the manuscript that addresses the points raised during the review process.

Please carefully read the reviewers' comments.

We look forward to receiving your revised manuscript.

Kind regards,

Dariusz Siudak, Ph.D., DSc.

Academic Editor

PLOS ONE

Journal Requirements:

"This research was supported by the ICT Division of Bangladesh [The grant number is 19FS32950]."

4. Please expand the acronym “ICT” (as indicated in your financial disclosure) so that it states the name of your funders in full.

Additional Editor Comments:

1. It seems that either formula (2) is incorrect or has been entered incorrectly.

2. A chart for the DSE stock index for the same period should be presented in Fig. 1 or separately.

3. It is hard to agree with the presented conclusions on page 17 (lines 337-380). The COVID-19 pandemic originated in December 2019 in China. Did the financial market in Bangladesh anticipate the negative impact of COVID-19 on its functioning? Or is it, however, the result of other macroeconomic factors, including systematic risk? The COVID-19 pandemic is considered a Black Swan event in the financial market, a completely unpredictable occurrence with broad implications for financial markets. In my opinion, this passage should be reformulated, attempting to find alternative explanations for changes in the Hierarchy coefficient.

Reviewers' comments:

Reviewer's Responses to Questions

**Comments to the Author**

1. Is the manuscript technically sound, and do the data support the conclusions?

Reviewer #1: Yes

Reviewer #2: No

Reviewer #3: Partly

2. Has the statistical analysis been performed appropriately and rigorously? 

Reviewer #1: Yes

Reviewer #2: No

Reviewer #3: Yes

3. Have the authors made all data underlying the findings in their manuscript fully available?

Reviewer #1: Yes

Reviewer #2: Yes

Reviewer #3: No

4. Is the manuscript presented in an intelligible fashion and written in standard English?

Reviewer #1: Yes

Reviewer #2: No

Reviewer #3: No

5. Review Comments to the Author

Reviewer #1: Recommendation: Major Revision

This paper investigates the hierarchical structure of Dhaka stocks’ financial networks. However, the execution is lacking number of fundamental issues.

What are your contributions? Better highlight novelty in the study. Better define motivations for the research.

Page 3, line 14. As far as I know, the Chinese stock market is still an emerging market compared to the U.S. Please ensure the expression is appropriate.

Introduction - Lack of appropriate critical appraisal of included studies to find current research gaps.

The choice of MST needs to be motivated. The authors merely list alternative models, such as random matrix theory, wavelet method, principal component analysis, and GARCH method. Instead, they should present a comprehensive summary of the advantages and disadvantages of each method.

PMFG preserves the same hierarchical structure as MST but offers more significant and richer information. Thus, the question arises: Why would one choose to utilize MST over PMFG. I suggest the author read this paper:

Di Matteo, T., Pozzi, F., & Aste, T. (2010). The use of dynamical networks to detect the hierarchical organization of financial market sectors. The European Physical Journal B, 73, 3-11.

The last waves of COVID-19 and the Russia-Ukraine conflict are two essential events for financial stability. The authors should cover these periods.

The DSE, such abbreviations, must be defined at their first mention there.

Line 289. The hub and the peripheral are measured by which of the centralities? The author should also show the hub ranking of all assets.

In Fig.1. The author should elaborate on why the peripheral company, AB Bank Limited, fell sharply in 2012, and Advance Chemical Industries Limited had a different trend compared to the other three companies during 2010 and 2011.

Figs. 4-7. The authors should supplement the U.S. market in figures as a contrast.

The conclusion section seems to rush to the end. The authors will have to demonstrate the impact and insights of the research. Clearly state your unique research contributions in the conclusion section.

Reviewer #2: Emerging markets are interesting, and network analysis can be useful for analyzing them. However, I don't believe that the article is in good shape for publication. The authors should restructure the article, avoiding redundancies and focusing more on results and analyses. The article contains a lot of information that obscures the central theme of the research.

Additionally, some statistical analyses deserve more attention. For me, it is not surprising that the distribution of sample correlations deviates from the Gaussian distribution.

I suggest the authors should consider revisiting the article's organization to avoid redundancies and provide a cohesive and easy-to-follow narrative."

Reviewer #3: The authors analyzed the Bangladesh stock market from the perspective of network hierarchy using the MST methodology. The overall structure of the paper is appropriate. However, it is necessary to reflect the following modifications.

1. Analyzing the Bangladesh stock market is meaningful for the local economy. However, from a global perspective, the importance of the Bangladesh stock market is relatively low compared to countries such as the United States, China, and Japan, for example. Therefore, the authors need to convince people of the implications of analyzing the Bangladesh stock market for financial markets around the world.

2. Why was the MST methodology used? What can we gain from analyzing the hierarchy of a network?

3. Why was normalized log-return used in equation (1)?

4. Explain equation (2) in more detail.

5. The format of the pictures does not match the journal, and the clarity is poor.

6. “which is useful in forecasting any upcoming big crisis” => Is it possible to predict risks? If possible, further explanation and analysis is needed to determine how possible this is.

7. "We find that the network hub and central nodes with a maximum degree and BC measure are mainly financial companies, more specifically the insurance companies, i.e., they play important roles in the Bangladeshi market." => What is the reason for these results? Or what do you think?

8. “We need analyzes on other emerging markets like Egypt, Thailand, Indonesia, etc., which will be left for further investigation.” => Please cite related existing literature.

9. “Which can be important to the respective authorities, investors, and policymakers for risk management and portfolio investment.” => Please provide more specific implications.

6. PLOS authors have the option to publish the peer review history of their article (what does this mean?). If published, this will include your full peer review and any attached files.

Reviewer #1: No

Reviewer #2: No

Reviewer #3: No

---

## [Author Response · Author response to Decision Letter 0]

13 Feb 2024

We have incorporated your suggestions into our revision. They were very helpful. Thank you.

---

## [Decision Letter · Decision Letter 1]

29 Feb 2024

PONE-D-23-35741R1Change in hierarchy of the financial networks: A study on firms of an emerging market in BangladeshPLOS ONE

Dear Dr. Nobi,

Thank you for submitting your manuscript to PLOS ONE. After careful consideration, we feel that it has merit but does not fully meet PLOS ONE’s publication criteria as it currently stands. Therefore, we invite you to submit a revised version of the manuscript that addresses the points raised during the review process.

We look forward to receiving your revised manuscript.

Kind regards,

Dariusz Siudak, Ph.D., DSc.

Academic Editor

PLOS ONE

Journal Requirements:

Reviewers' comments:

Reviewer's Responses to Questions

**Comments to the Author**

1. If the authors have adequately addressed your comments raised in a previous round of review and you feel that this manuscript is now acceptable for publication, you may indicate that here to bypass the “Comments to the Author” section, enter your conflict of interest statement in the “Confidential to Editor” section, and submit your "Accept" recommendation.

Reviewer #1: All comments have been addressed

Reviewer #2: All comments have been addressed

Reviewer #3: (No Response)

2. Is the manuscript technically sound, and do the data support the conclusions?

Reviewer #1: (No Response)

Reviewer #2: Yes

Reviewer #3: (No Response)

3. Has the statistical analysis been performed appropriately and rigorously? 

Reviewer #1: Yes

Reviewer #2: Yes

Reviewer #3: (No Response)

4. Have the authors made all data underlying the findings in their manuscript fully available?

Reviewer #1: Yes

Reviewer #2: Yes

Reviewer #3: (No Response)

5. Is the manuscript presented in an intelligible fashion and written in standard English?

Reviewer #1: Yes

Reviewer #2: Yes

Reviewer #3: (No Response)

6. Review Comments to the Author

Reviewer #1: (No Response)

Reviewer #2: The authors have made revisions following the recommendations of the reviewers. It appears more suitable for publication now.

Suggestion to the authors

In the introduction: “The topological...US and China [3] “ -- “ In particular, the topological...US and China [3]”.

Reviewer #3: Query 7: "We find that the network hub and central nodes with a maximum degree and BC measure are mainly financial companies, more specifically the insurance companies, i.e., they play important roles in the Bangladeshi market." => What is the reason for these results? Or what do you think?

Response: The insurance sector plays a vital role in the Bangladeshi market. According to PwC [8], the insurance sector has the potential to make a positive contribution to the overall standard of living of the country's populace. The sector can help reduce the burden on the overall spending on healthcare. The insurance sector can also help manage risks in an economy at a micro level, safeguarding households and companies from a myriad of risks [8].

Please include the response for the Q7 longer in the manuscript. Other modifications are satisfied.

7. PLOS authors have the option to publish the peer review history of their article (what does this mean?). If published, this will include your full peer review and any attached files.

Reviewer #1: **Yes: **Longfeng Zhao

Reviewer #2: No

Reviewer #3: No

---

## [Author Response · Author response to Decision Letter 1]

8 Mar 2024

Reviewer 2:

Suggestion 1: In the introduction: “The topological...US and China [3] “ -- “ In particular, the topological...US and China [3]”.

Response: We have modified this line according to your suggestion. Thank you very much for your valuable advice. 

Reviewer 3:

Query 1: Query 7: "We find that the network hub and central nodes with a maximum degree and BC measure are mainly financial companies, more specifically the insurance companies, i.e., they play important roles in the Bangladeshi market." => What is the reason for these results? Or what do you think?

Response: The insurance sector plays a vital role in the Bangladeshi market. According to PwC [8], the insurance sector has the potential to make a positive contribution to the overall standard of living of the country's populace. The sector can help reduce the burden on the overall spending on healthcare. The insurance sector can also help manage risks in an economy at a micro level, safeguarding households and companies from a myriad of risks [8].

Please include the response for the Q7 longer in the manuscript. Other modifications are satisfied.

Response: We have enlarged the response for the Q7 of the previous revision as follows:

“The insurance sector plays a vital role in the Bangladeshi market. For a poor country like Bangladesh, the influence of the insurance company is remarkable since it has been crucial in the lives of mass people and the related aspects those are concerned about. According to PwC [1], the insurance sector has the potential to make a positive contribution to the overall standard of living of the country's populace. The sector can help reduce the burden on the overall spending on healthcare. The insurance sector can also help manage risks in an economy at a micro level, safeguarding households and companies from a myriad of risks [1]. 

Although the influence of a company depends on various factors such as P/E (Price-Earnings ratio), EPS (Earnings Per Share), ROA (Return on Assets), SZ (Firm’s Size), etc., the average P/E ratio of sample insurance companies in DSE is below 20 (except for 2011) [2], indicating a favorable investment opportunity for small-scale investors. It is also seen that the ROA and EPS of some selected insurance companies have significant positive impacts on market movement. These are a few possible factors why insurance companies are influential in DSE.”

We have included this response in two different sections of the manuscript, in the “Central companies” subsection and “Conclusion” based on its relevancy.

---

## [Decision Letter · Decision Letter 2]

22 Mar 2024

Change in hierarchy of the financial networks: A study on firms of an emerging market in Bangladesh

PONE-D-23-35741R2

Dear Dr. Nobi,

We’re pleased to inform you that your manuscript has been judged scientifically suitable for publication and will be formally accepted for publication once it meets all outstanding technical requirements.

Kind regards,

Dariusz Siudak, Ph.D., DSc.

Academic Editor

PLOS ONE

Additional Editor Comments (optional):

Reviewers' comments:

Reviewer's Responses to Questions

**Comments to the Author**

1. If the authors have adequately addressed your comments raised in a previous round of review and you feel that this manuscript is now acceptable for publication, you may indicate that here to bypass the “Comments to the Author” section, enter your conflict of interest statement in the “Confidential to Editor” section, and submit your "Accept" recommendation.

Reviewer #2: All comments have been addressed

Reviewer #3: All comments have been addressed

2. Is the manuscript technically sound, and do the data support the conclusions?

Reviewer #2: Yes

Reviewer #3: Yes

3. Has the statistical analysis been performed appropriately and rigorously? 

Reviewer #2: Yes

Reviewer #3: Yes

4. Have the authors made all data underlying the findings in their manuscript fully available?

Reviewer #2: Yes

Reviewer #3: Yes

5. Is the manuscript presented in an intelligible fashion and written in standard English?

Reviewer #2: Yes

Reviewer #3: Yes

6. Review Comments to the Author

Reviewer #2: In my opinion the authors have adequately addressed all comments raised by the reviewers.

In the current version I suggest publishing the article.

Reviewer #3: I have confirmed the revision and am satisfied. I would like to thank the author for his hard work..

7. PLOS authors have the option to publish the peer review history of their article (what does this mean?). If published, this will include your full peer review and any attached files.

Reviewer #2: No

Reviewer #3: No

---

## [Editor Report · Acceptance letter]

21 May 2024

PONE-D-23-35741R2 

PLOS ONE

Dear Dr. Nobi, 

I'm pleased to inform you that your manuscript has been deemed suitable for publication in PLOS ONE. Congratulations! Your manuscript is now being handed over to our production team.

Kind regards, 

on behalf of

Dr. Dariusz Siudak 

Academic Editor

PLOS ONE